# Molecular Analysis of Indole and Skatole Decomposition Metabolism in *Acinetobacter piscicola* p38 Utilizing Biochemical and Omics Approaches

**DOI:** 10.3390/microorganisms12091792

**Published:** 2024-08-29

**Authors:** Zhonghao Wang, Jiajin Sun, Pu Yang, Wanjun Zhang, Yihong Jiang, Qiang Liu, Yunqi Yang, Ruirong Hao, Gang Guo, Wenjie Huo, Qiang Zhang, Qinghong Li

**Affiliations:** 1College of Animal Science, Shanxi Agricultural University, Taigu 030800, China; zhonghaowang@stu.sxau.edu.cn (Z.W.); jiajinsun@stu.sxau.edu.cn (J.S.); yp11112024@163.com (P.Y.); zhangwanjun0424@163.com (W.Z.); yihongjiang@stu.sxau.edu.cn (Y.J.); qiangliu@stu.sxau.edu.cn (Q.L.); yangyunqi@stu.sxau.edu.cn (Y.Y.); hrr823229@126.com (R.H.); guosteel1984@163.com (G.G.); huohuo-1982@163.com (W.H.); 2College of Resources and Environment, Shanxi Agricultural University, Taigu 030800, China; zhangqiang0351@163.com

**Keywords:** indole, skatole, oxygenase, gene cluster, prokaryotic transcriptome, biodegradation

## Abstract

Indole and skatole (3-methylindole, C_9_H_9_N) are common nitrogen-containing heterocyclic pollutants found in waste, wastewater treatment plants, and public restrooms and are the most notorious compounds in animal feces. Biodegradation was considered a feasible method for the removal of indole and skatole, but a comprehensive understanding of the metabolic pathways under both aerobic and anaerobic conditions was lacking, and the functional genes responsible for skatole biodegradation remained a mystery. Through metagenomic and gene cluster functional analysis, *Acinetobacter piscicola* p38 (NCBI: CP167896), *genes 1650* (styrene monooxygenase: ACDW34_08180), and *1687* (styrene monooxygenase: ACDW34_08350) were identified as having the potential to degrade indole and skatole. The heterologous expression results demonstrate that the genes 1650 and 1651 (flavin reductase: ACDW34_08185), when combined, are capable of degrading indole, while the genes 1687 and 1688 (flavin reductase: ACDW34_08355), in combination, can degrade indole as well as skatole. These reactions necessitate the involvement of flavin reductase and NAD(P)H to catalyze the oxygenation process. This work aimed to provide new experimental evidence for the biodegradation of indole and skatole. This study offered new insights into our understanding of skatole degradation. The *Acinetobacter_piscicola* p38 strain provided an effective bacterial resource for the bioremediation of fecal indole and skatole.

## 1. Introduction

Indole and skatole (3-methylindole, chemical formula C_9_H_9_N) are typical nitrogen-containing heterocyclic pollutants that are ubiquitous in environments such as waste treatment plants, wastewater treatment facilities, and public restrooms [1,2]. They are notorious not only for their strong odor but also for the potential health risks they pose, which have garnered increasing attention [3]. These compounds are particularly concentrated in animal feces, presenting a significant challenge to environmental management in agriculture and animal husbandry [4]. Long-term exposure to these compounds may trigger a range of health issues, including respiratory diseases, skin irritation, and neurological disorders. The severity of these issues has prompted the need for effective measures to reduce their concentrations in the environment [5,6].

In this context, biodegradation has emerged as an environmentally friendly method for the removal of indole and skatole [7,8]. Biodegradation harnesses the metabolic activities of microorganisms to convert these harmful substances into harmless or less toxic forms, thereby mitigating their impact on the environment and health [9]. However, the current understanding of the metabolic pathways of these compounds under aerobic and anaerobic conditions is not comprehensive, which limits our ability to develop more efficient biodegradation technologies [10]. In particular, the key functional genes and metabolic enzymes responsible for the biodegradation of skatole remain largely unknown. Identifying and characterizing these genes and enzymes is crucial for understanding how microorganisms degrade these pollutants and are key to developing new biotechnologies. With the advancement of molecular biology techniques, we now have more opportunities to delve into the biodegradation mechanisms of these pollutants. Through genomics and transcriptomics, researchers can identify and characterize key genes and enzymes involved in the degradation process [11]. These studies not only help us understand how microorganisms adapt to and metabolize these complex organic compounds but also provide a scientific basis for developing new biocatalysts and bioremediation strategies.

Indole and its derivatives, including skatole among the N-heterocyclic compounds, are a class of organic compounds widely distributed in the environment [12]. As shown in Figure 1, the degradation of these compounds in nature mainly relies on the action of microorganisms, and their biodegradation processes can be divided into anaerobic and aerobic pathways [13,14]. Anaerobic degradation typically involves the participation of acyl-CoA, a process that converts indole derivatives into forms more easily metabolized by microorganisms under anaerobic conditions [13]. This pathway involves acylation reactions with coenzyme A, transferring the acyl group from coenzyme A to the compound to form acylated products. In contrast, aerobic degradation primarily depends on the action of oxygenases, enzymes that catalyze the direct reaction of molecular oxygen with the substrate, initiating a series of oxidative reactions that promote the degradation process. Oxygenases are enzymes that can directly bind molecular oxygen to the substrate, catalyzing the initial oxidation reaction in the aerobic degradation process, activating the substrate molecules, and paving the way for subsequent metabolic pathways [15,16]. This step is crucial for breaking the heterocyclic structure of indole and its derivatives, transforming compounds that are otherwise difficult to metabolize into forms more readily utilized by microorganisms. Studies have shown that various bacteria, such as Acinetobacter baumannii [17], Paraburkholderia phytofirmans [18], Enterobacter soli [19], Pseudomonas putida [20], and Caballeronia glathei [14], have the ability to degrade N-heterocyclic compounds, and oxygenases have been confirmed to play a key initial reaction role in the biodegradation of N-heterocyclic compounds. Therefore, an in-depth study of the role of oxygenases and acyl-CoA in the degradation of N-heterocyclic compounds is significant for understanding the environmental behavior of these compounds and developing effective bioremediation strategies.

The microbial degradation of indole and its derivatives plays a crucial role in environmental management and biotechnological applications, with oxygenases such as YcnE, IifC, and IndA playing essential roles in the initial stages of indole degradation [12,21,22]. YcnE, derived from Enterococcus hirae GDIAS-5, activates the indole molecule through a two-component indole oxygenase system, initiating its degradation process [21]. IifC, isolated from Burkholderia sp. IDO3, provides a new perspective for understanding the metabolic pathways of indole through its characteristics and functional gene analysis [12]. IndA, found in a *Cupriavidus* sp. strain, has had its specific mechanism in indole biotransformation revealed by Y. Qu and colleagues in their 2017 study [22]. The discovery of these oxygenases not only enhances our understanding of indole metabolic pathways but also provides a molecular basis for developing microbial degradation technologies to address environmental pollutants containing indole.

Various microbial strains, including Acinetobacter oleivorans AO-06 [23], Lactobacillus brevis 1.12 [24], Acinetobacter toweneri NTA1-2A [25], and *Cupriavidus* sp. strain KK10 [26], have been proven to effectively degrade skatole. Moreover, research on the biodegradation of skatole has revealed the degradation capabilities of multiple microbial strains under different environmental conditions. For instance, Pseudomonas putida LPC24 can degrade 3-methylindole under oxygen-limited conditions [27], while Pseudomonas aeruginosa Gs carries out degradation under aerobic conditions [28]. *Cupriavidus* sp. strain KK10 has demonstrated the ability to convert 3-methylindole into ring-cleavage products under aerobic conditions [26]. Additionally, marine anaerobic microorganisms have also been shown to degrade such compounds [29]. Although the specific genes involved have not yet been reported, potential aerobic and anaerobic degradation mechanisms have been indicated. These findings highlight the diversity and complexity of microorganisms in degrading environmental pollutants and point the way for future research directions.

The research direction of this study aims to screen and optimize microbial strains with efficient degradation capabilities for indole and skatole and to delve into the catalytic characteristics of oxygenases/acyl-CoA and other key enzymes at the molecular level during the degradation process. By revealing the key enzymes and reaction steps in the metabolic pathway, we can provide more effective strategies and methods for environmental protection and pollution control. The significance of these comprehensive studies lies in deepening our understanding of biodegradation mechanisms, promoting the development of eco-friendly biotechnologies, and providing innovative strategies and solutions for environmental pollution management, contributing to ecosystem management and biodiversity conservation.

## 2. Materials and Methods

### 2.1. Materials

Agar (CAS 9002-18-0 Agar powder, strength 1400), yeast extract (Oxoid LP0021), peptone (Oxoid LP0042B), sodium chloride, petri dishes, skatole (3-methylindole), indole, KH_2_PO_4_, K_2_HPO_4_, MgSO_4_, anhydrous ethanol, saline solution, LiChrospher^®^ RP-18 HPLC column 5 μm particle size, L × I.D. 25 cm × 4.6 mm (Sigma-Aldrich (Shanghai) Trading Co. Ltd., Shanghai, China), acetonitrile (chromatography grade), methanol (chromatography grade), Tris buffer, 2-Methylindole, 0.2 μm organic filter membrane, syringes, microwave oven.

### 2.2. Determination of Skatole by High-Performance Liquid Chromatography (HPLC)

Establishment of indole and skatole standard curve: prepare skatole solutions at concentrations of 100 μg/mL, 50 μg/mL, 10 μg/mL, 5 μg/mL, 2.5 μg/mL, 1 μg/mL, and 0.5 μg/mL (100 μL standard sample + 500 μL 0.05 M Tris buffer + 400 μL chromatographic acetonitrile); mobile phase: water: acetonitrile: methanol = 52:40:8; temperature: 35 °C; Flow rate: 1 mL/min for 20 min; excitation wavelength 280 nm, emission wavelength 352 nm; Injection volume 20 μL.

### 2.3. Screening of Skatole Biodegrading Bacteria Using Skatole as the Sole Carbon Source

Medium formulation: LB liquid medium: peptone 10 g, NaCl 5 g, yeast extract 5 g, distilled water up to 1 L. Inorganic salt medium: KH_2_PO_4_ 0.5 g, K_2_HPO_4_ 1.5 g, MgSO_4_ 0.5 g, (NH_4_)_2_SO_4_ 1.5 g, distilled water up to 1 L, pH (no adjustment needed). Enrichment and domestication medium: on the basis of inorganic salt medium, gradient increase (10 to 100 mg/L) of skatole. Separation medium: add skatole to the enrichment and domestication culture liquid to make the final concentration 50 mg/L, agar 20 g. Skatole mother liquid: dissolve 1 g of skatole in 25 mL of anhydrous ethanol to make a 40 mg/mL mother liquid, and store at 4 °C for later use.

Detailed experimental steps:

Collection and processing of fecal samples: 2 kg of fresh feces was collected from the pig, chicken, cattle, and sheep experimental centers of the Animal Science and Technology Experiment Station of Shanxi Agricultural University. The feces from different sources were divided into two portions (1 kg each): one portion was buried in a shallow soil layer, and the other was piled on the ground surface. Preliminary screening of microorganisms: After one month, feces with different treatment methods were collected. The feces from the same animal were mixed evenly, 200 g was taken and soaked in 500 mL of saline solution, and it was placed in a shaker for 1 h (37 °C, 150 r/min) to extract microorganisms. Step 1: 10 mL of the treated liquid was transferred to the enrichment and domestication medium (skatole: 10 mg/L), with two replicates per group, and incubated at 37 °C for 2 days. Step 2: after 2 days, 10 mL of the treated liquid was transferred to a new enrichment and domestication medium (skatole: 20 mg/L) and incubated at 37 °C for 2 days. Step 3: after 2 days, 10 mL of the treated liquid was transferred to a new enrichment and domestication medium (skatole: 50 mg/L) and incubated at 37 °C for 2 days. Step 4: after 2 days, 10 mL of the treated liquid was transferred to a new enrichment and domestication medium (skatole: 100 mg/L) and incubated at 37 °C for 2 days. Step 5: 80 separation media (skatole: 50 mg/L) were prepared, and the liquid from step four was spread on the separation medium using the dilution spread plate method and incubated at 37 °C for 2 days. Step 6: 10 colonies of different morphologies were picked from the culture of step five, selected based on colony morphology and size. Single colonies were transferred to 5 mL of domestication medium (skatole: 100 mg/L) and incubated at 37 °C at 180 rpm for 1 day (this step screened for 23 cultures with thickened liquid). Step 7: the enriched colonies from step six were cultivated using the streak plate method, with skatole at 50 mg/L, and incubated at 37 °C for 2 days. Step 8: single colonies from the culture of step seven were transferred into 100 mL of enrichment medium (skatole: 100 mg/L), and the culture liquid was collected to determine the content of skatole after 24 h. In total, 1 mL of culture liquid was taken at 12,000 rpm for 20 min. An amount of 100 μg of the supernatant was taken, 400 μL of chromatographic acetonitrile was added, followed by 500 μL of Tris buffer containing 10 μg/mL of 2-methylindole (internal standard), mixed well, and filtered through a 0.22 μm organic filter membrane. Step 9: the top 2 bacterial strains were selected, and the types of strains were identified. Step 10: based on the identified types of strains, the effects of different times, temperatures, and pH on the degradation rate of skatole were investigated. Step 11: a skatole degradation experiment was conducted with the best *Acinetobacter_piscicola* p38 strain (medium: 100 mg/L skatole, KH_2_PO_4_ 0.5 g, K_2_HPO_4_ 1.5 g, MgSO_4_ 0.5 g, (NH_4_)_2_SO_4_ 1.5 g, distilled water up to 1 L), and samples were sent for chemical analysis. Step 12: based on the experimental results, the best *Acinetobacter_piscicola* p38 (NCBI: CP167896) strain was selected for whole genome sequencing. Step 13: cultivate *Acinetobacter_piscicola* p38 in an LB medium containing 50 mg/L indole to measure its indole degradation capability. Step 14: according to the whole genome sequencing results, the *Acinetobacter_piscicola* p38 strain was cultivated in the LB medium (50 mg/L skatole; 50 mg/L indole) for 3 h to collect bacterial sediment, and prokaryotic transcriptome sequencing was performed.

### 2.4. Bacterial Identification and Phylogenetic Analysis

A single colony was picked from the solid culture medium and enriched for 24 h to cultivate. The bacterial sediment was collected. It was preserved with dry ice and shipped to Majorbio Medical Technologies Co., Ltd. in Shanghai, China, for 16S rRNA gene sequencing. The 16S rRNA results were used to perform a BLAST search on the NCBI database to download the matching sequences. Then, a phylogenetic tree analysis was conducted using the maximum likelihood method on the website https://meinverse.cn (accessed on 23 March 2024).

### 2.5. Whole Genome Sample Preparation and Analysis

A single colony was selected from the solid culture medium, and an enrichment culture was performed for 24 h. The culture was then centrifuged at 12,000 rpm for 20 min to collect the bacterial pellet. The pellet was preserved with dry ice and shipped to Majorbio Medical Technologies Co., Ltd. in Shanghai, China, for 16S rRNA gene sequencing and whole genome sequencing. Majorbio analyzed the sequencing results.

### 2.6. Prokaryotic Transcriptome Sample Preparation and Analysis

A single colony was selected from the solid culture medium, and an LB medium culture was performed for 24 h. Then, 1 mL was transferred to a new culture medium and cultivated for an additional 6 h to collect the bacteria. Three types of media were used for this process: LB medium, 50 mg/L skatole LB medium, and 50 mg/L indole LB medium, with three replicates for each group. The cultures were centrifuged at 12,000 rpm for 20 min to collect the bacterial pellet. The pellet was preserved with dry ice and shipped to Majorbio Medical Technologies Co., Ltd. in Shanghai, China, for 16S rRNA gene sequencing and transcriptome sequencing. Majorbio analyzed the sequencing results.

### 2.7. Heterologous Expression of Genes and Degradation of Indole and Skatole

The genome of the *Acinetobacter_piscicola* p38 strain was extracted, and the oxygenase genes and flavin reductase were cloned using PCR technology (primer details are attached 1). Homologous recombination technology was employed to assemble the target genes with the Pet28a plasmid (containing a 6×His tag) and then transferred into BL21(DE3). Positive clones were selected for sequencing. The target heterologous expression was carried out using a TB medium, with an OD600 between 0.6 and 0.8, an IPTG concentration of 0.1 mM, at a temperature of 16 °C, and a shaking speed of 100 rpm/min for induction for 24 h. The cells were collected and washed twice with PBS. For whole-cell biotransformation, the collected E. coli were transferred to fresh LB medium (with the addition of 50 mg/L indole and skatole), cultured at 37 °C and 150 rpm/min on a shaking bed for 12 h, and then the supernatant was collected to measure the content of indole and skatole.

### 2.8. Enzyme Analysis and Detection of Reaction Products

The crude enzymes of gene1650 (styrene monooxygenase: ACDW34_08180), gene1651 (flavin reductase: ACDW34_08185), gene1687 (styrene monooxygenase: ACDW34_08350), and gene1688 (flavin reductase: ACDW34_08355) after induced expression were extracted and the target proteins were collected using a nickel column. Use the BCA Protein Assay Kit to determine the protein concentration, and then adjust the concentration of the single protein to 10 µg/mL in the subsequent reaction system. After purification, the proteins were reacted separately with 60 mmol/L indole and skatole, containing 10 µM FAD and 100 µM NAD(P)H, cultured at 37 °C and 150 rpm/min on a shaking bed for 12 h, and then the supernatant was collected to measure the concentration of indole and skatole. Different concentrations of indole and skatole (ranging from 10 to 3000 µM) are prepared, each containing 10 µM FAD and 100 µM NAD(P)H. The reaction is initiated by the addition of 5 µg of oxygenase and 5 µg of flavin reductase. The mixture is incubated at room temperature for 20 min, and every 4 min, 100 µL of the reaction mixture is removed and mixed with 400 µL of acetonitrile to quench the reaction. Subsequently, 500 µL of 0.05 M Tris-HCl buffer is added.

### 2.9. Protein Phylogenetic Tree Analysis

Compare the amino acid sequences of gene1650 and gene1687 using NCBI-Blast and UniProt-Blast, respectively, and collect the homologous protein sequences (Appendix A). Evolutionary analyses were conducted in MEGA11. The evolutionary history was inferred using the Neighbor-Joining method. The optimal tree is shown. The percentage of replicate trees in which the associated taxa clustered together in the bootstrap test (1000 replicates) are shown next to the branches. The tree is drawn to scale, with branch lengths in the same units as those of the evolutionary distances used to infer the phylogenetic tree. The evolutionary distances were computed using the Poisson correction method and are in the units of the number of amino acid substitutions per site. This analysis involved 7 amino acid sequences. All ambiguous positions were removed for each sequence pair (pairwise deletion option). There was a total of 452 positions in the final dataset.

### 2.10. Data Analysis

Majorbio Medical Technologies Co., Ltd. conducted a bioinformatics analysis of the *Acinetobacter_piscicola* p38 strain and its prokaryotic transcriptome. The metabolomic analysis of the culture medium was utilized by Qingdao STD Standard Testing Co., Ltd (Qingdao, China). Routine data in this paper were recorded using Excel (Microsoft® Excel® 2019MSO, Microsoft Corporation, Redmond, Washington, DC, USA), and figures were drawn using KingDrawPc_V3.0.2.20 (Qingdao Qingyuan Precision Agriculture Technology Co., Ltd., Qingdao, China), GraphPad Prism 9.0 (GraphPad Software, San Diego, CA, USA), and Adobe Illustrator 2022-26.0 (Adobe Systems, San Jose, CA, USA) for graphic illustration. In the determination of indole and skatole degradation, we employed the *t*-test in IBM SPSS Statistics 26.0.0 (IBM Corporation, Armonk, New York, NY, USA), with a statistically significant difference indicated by a *p*-value of less than 0.05.

## 3. Results and Discussion

### 3.1. Screening of Skatole-Degrading Bacteria

In this study, we successfully screened and cultivated 40 individual colonies through enrichment culture of fecal samples and dilution plating methods. In the enrichment culture medium containing 100 mg/L, after 24 h of cultivation, 23 colonies successfully grew (Table 1), especially colonies pig35 and 38, which demonstrated the ability to completely degrade skatole within 24 h, and this result was consistent in repeated experiments. More notably, these colonies achieved a 100% degradation rate within 6 h in the culture medium of the same concentration, indicating their extremely high degradation efficiency. Through 16S RNA analysis, we identified the species of pig35 and 38. Their 16s RNA sequences were consistent and confirmed by the NCBI database as *Acinetobacter_piscicola*; thus, in this study, they are named *Acinetobacter_piscicola* p38. The phylogenetic tree results are shown in Figure 2a.

Skatole, as a compound of significant interest, has seen remarkable progress in its biodegradation research. In this study, the *Acinetobacter_piscicola* p38 strain demonstrated exceptional degradation efficiency under conditions where skatole was the sole carbon source, being able to completely degrade 100 mg/L of skatole within just 6 h. This degradation rate far exceeds that reported for other strains in other studies, such as Acinetobacter oleivorans AO-06, which can completely degrade up to 100 mg/L of 3MI in 48 h [23], Lactobacillus brevis 1.12, which has a 65% degradation rate for 1.0 mg/L of 3MI in 120 h [24], Acinetobacter toweneri NTA1-2A, which has a degradation rate higher than 97% for 65.58 mg/L of 3MI in 72 h [25], and *Cupriavidus* sp. strain KK10, which achieves a 100% degradation rate for 100 mg/L of 3MI in 24 h [26]. Compared to these strains, *Acinetobacter_piscicola* p38 shows a faster degradation speed and higher efficiency.

The *Acinetobacter_piscicola* p38 strain showed high biodegradation capability for skatole in our study, but its degradation efficiency was affected by growth temperature, pH value, and the type of culture medium. The degradation efficiency peaked at 37 °C (Figure 2c), and the best efficiency was observed in the culture medium with a pH of 7 (Figure 2d). As depicted in Figure 2b, our research has uncovered that the *Acinetobacter_piscicola* p38 strain possesses the ability to metabolize skatole as a carbon source through a catabolic process. In the enrichment medium, although we only provided the essential inorganic elements such as nitrogen, oxygen, sulfur, and phosphorus, without the addition of a specific carbon source, the p38 strain was still able to effectively utilize skatole for growth. This further confirms the p38 strain’s capability to use skatole from the environment as a carbon source. The degradation of skatole by p38 not only demonstrates its adaptability in catabolic metabolism but also substantiates that the degradation of skatole by *Acinetobacter_piscicola* p38 is indeed a catabolic process. Notably, in the 100 mg/L skatole-enriched culture medium, *Acinetobacter_piscicola* p38 achieved a 100% biodegradation rate in just 6 h. In contrast, in the 100 mg/L LB medium, although a 100% degradation efficiency was eventually achieved, the process required an extended period of 24 h to complete (Figure 2b). This reduction in degradation efficiency may be due to competition between the rich carbon source in the LB medium and skatole, affecting the biological utilization of skatole by *Acinetobacter_piscicola* p38. These findings indicate that by optimizing the culture conditions, the degradation efficiency of *Acinetobacter_piscicola* p38 for skatole can be significantly improved, providing important guidance for bioremediation strategies. In summary, this study not only reveals the high-efficiency characteristics of *Acinetobacter_piscicola* p38 in skatole degradation but also compares it with strains from other studies, highlighting its potential for application in the field of bioremediation.

### 3.2. Metabolite Detection of Metabolism with Skatole as the Sole Carbon Source

In this study, we conducted a detailed mass spectrometry analysis of the degradation capabilities of *Acinetobacter_piscicola* p38 in a 100 mg/L enriched culture medium. The experimental results showed that during the 4 h cultivation process, although complete degradation was not achieved, the presence of skatole in the culture medium was still detectable, indicating that the biodegradation process had begun and may have produced some intermediate metabolites. Through mass spectrometry detection, we identified a variety of potential degradation intermediates, including indole, 2-aminobenzene acetone, 3-methoxyindole, o-hydroxybenzoic acid, (E)-3-(4-hydroxyphenyl)acrolein, 3-indole aldehyde, 2-hydroxyquinoline, quinoline, L-tyrosine, L-phenylalanine, N-(2,4-dimethylphenyl)formamide, L-dopa, etc. The chemical formulas of these compounds are mainly C8H9N and C9H9NO, most of which contain oxygen atoms (Appendix A). This finding supports the hypothesis that skatole may be biodegraded through an aerobic degradation pathway.

In addition, we also detected common metabolic products of indole in the culture medium, such as anthranilic acid and benzoic acid (Figure 2e) [30], which further confirms the potential of *Acinetobacter_piscicola* p38 to degrade indole. These results not only reveal the potential metabolic pathways of *Acinetobacter_piscicola* p38 in degrading skatole but also suggest the potential of this strain to degrade indole.

To verify this hypothesis, we plan to conduct the following experiments: First, *Acinetobacter_piscicola* p38 will be cultivated in an LB medium (containing 100 mg/L indole) to assess its degradation capabilities. The experimental results show that *Acinetobacter_piscicola* p38 has the ability to degrade indole. This helps to further understand the degradation of indole by *Acinetobacter_piscicola* p38 and provides a scientific basis for developing new bioremediation strategies.

### 3.3. Whole Genome and Potential Gene Analysis of Acinetobacter_Piscicola p38

In this study, as shown in Figure 3a, we performed whole-genome sequencing of *Acinetobacter_piscicola* p38 and conducted in-depth bioinformatics analysis using databases such as NR, Swiss-prot, Pfam, COG, GO, and KEGG. The analysis revealed that the p38 strain has a complete biological metabolic function, especially showing significant gene participation in the biodegradation and metabolism of xenobiotics. Through KEGG bioinformatics analysis, we identified 89 genes involved in the biodegradation and metabolism of xenobiotics. These genes are involved in 18 different metabolic pathways, including 38 genes involved in the benzoate degradation process, as shown in Figure 3b,c. A key step in the benzoate degradation process is the biodegradation of catechol, which is not only crucial for the biodegradation of skatole but also provides a carbon source for bacteria [31].

Furthermore, we conducted bioinformatics analysis of genes related to oxygenases and acyl-CoA in the *Acinetobacter_piscicola* p38 strain. The results found that there are 44 and 33 genes related to these two enzymes, respectively, as shown in Figure 3d (Appendix A). The existence of these genes indicates that the p38 strain has the ability to oxidize benzene ring compounds into intermediate products that can be further metabolized.

Integrating the above data analysis, we can infer that in the genome of *Acinetobacter_piscicola* p38, there are multiple genes related to benzene ring oxygenases and benzoate degradation, which may be closely related to the strain’s ability to biodegrade indole and skatole. The identification of these genes provides us with important information for further research and optimization of the strain’s biodegradation pathways and provides a molecular basis for the development of bioremediation technology based on the p38 strain.

### 3.4. Prokaryotic Transcriptome Analysis of Acinetobacter_Piscicola p38

This study delved into the biodegradation capabilities of the *Acinetobacter_piscicola* p38 strain for two typical nitrogen-containing heterocyclic pollutants: skatole and indole. By cultivating the strain in an LB medium with specific concentrations of these pollutants, we not only verified its efficient degradation capabilities but also revealed the underlying molecular mechanisms through prokaryotic transcriptome analysis. Initially, *Acinetobacter_piscicola* p38 was cultivated in two different media: one with 50 mg/L skatole added to the LB medium and the other with 50 mg/L indole added. The control group used a regular LB medium. After a 6 h cultivation period, the strains were subjected to cell lysis, and the cellular proteins were collected to assess their degradation capabilities. The results showed that compared to the regular LB medium, the cellular proteins from the pollutant-containing media exhibited significant pollutant removal capabilities, as shown in Figure 4a. This phenomenon clearly indicated that the presence of pollutants significantly enhanced the expression of related degradation genes.

To further understand this phenomenon, we conducted a prokaryotic transcriptome analysis of these strains. For the strain cultivated with indole, 1231 differential genes were found, with 633 genes upregulated and 598 genes downregulated, as shown in Figure 4b. A total of 841 differential genes were identified in the *Acinetobacter_piscicola* p38 strain cultivated with skatole, with 420 genes upregulated and 421 genes downregulated, as shown in Figure 4c. A considerable portion of these differential genes were enriched in metabolism-related KEGG pathways, particularly in the metabolism of xenobiotics. Specifically, 18 upregulated genes (Appendix A) in the skatole medium and 14 (Appendix A) in the indole medium were enriched in the benzoate degradation and catechol degradation (map00362) pathways, revealing the ability of *Acinetobacter_piscicola* p38 to degrade skatole and indole into catechol.

Further analysis also revealed that under both pollutant cultivation conditions, a series of specific genes (*gene2145* to *gene2151*) were involved in the biotransformation of catechol, enabling it to enter the tricarboxylic acid cycle, providing a carbon source for the bacteria and supporting Ma Q and others’ research findings [31]. This was visually presented in Figure 4d, clearly illustrating the process of material transformation. Additionally, an interesting phenomenon was observed: the crude enzyme extracted from the skatole medium had the ability to degrade indole, while the crude enzyme from the indole medium could not degrade skatole. This finding suggests that there may be degradation enzymes specific to skatole in the *Acinetobacter_piscicola* p38 strain.

Based on the existing research findings and observed phenomena, we plan to conduct an in-depth analysis of the oxygenase and acyl-CoA-related genes in the prokaryotic transcriptome. First, we will design experiments based on our hypotheses to compare the transcriptome expression differences of A. piscicola p38 under indole and skatole pollution conditions with the non-polluted control group, as shown in Table 2 and Table 3. After collecting data through RNA sequencing technology, we will use bioinformatics tools to screen for oxygenase and acyl-CoA genes with significant differential expression. Other data is in Appendix A.

### 3.5. Heterologous Expression Analysis of Oxygenases and Acyl-Coenzyme a in Acinetobacter_Piscicola p38

In this study, we focused on two key oxygenase genes, gene1650 and gene1687. Heterologous expression results show consistency with the predicted size, as shown in Figure 5a–c, selected from the *Acinetobacter_piscicola* p38 strain (Appendix A). Through heterologous expression and whole-cell biotransformation analysis, we found that both oxygenases could convert indole into indigo, as shown in Figure 5d,g. Particularly, gene1687 showed a higher degradation efficiency than gene1650 in liquid chromatography measurements and also had the ability to degrade skatole, which was verified in the whole-cell biotransformation experiment.

However, when attempting to directly react the purified gene1650 and gene1687 proteins with indole and skatole, as shown in Figure 5e,f, we did not observe the expected degradation phenomenon. This suggests that the activity of oxygenases may depend on specific auxiliary factors [14,21]. To explore this possibility, we heterologously expressed the flavin reductase genes gene1651 and gene1688 related to these oxygenases and successfully purified the corresponding proteins, as shown in Figure 5e,f.

The experimental results were encouraging: with the addition of coenzyme FAD and reduced coenzyme NAD(P)H to the reaction system, the combination of gene1650 and gene1651 successfully degraded indole. Similarly, the combination of gene1687 and gene1688, under conditions with added FAD and NAD(P)H, not only efficiently degraded indole but also degraded skatole, further explaining why the skatole-induced *Acinetobacter_piscicola* p38 strain gene expression could degrade indole. These findings confirm the importance of auxiliary factors in the activity of oxygenases and provide new strategies for further optimizing the degradation efficiency of these enzymes. Additionally, as shown in Figure 6a, the evolutionary tree comparison analysis of gene1650 and gene1687 revealed homology with styrene monooxygenase, which is very important because the oxygenase requires the oxidoreductase encoded by the corresponding gene in the gene cluster to facilitate its action (as depicted in Figure 6b), further underscoring the necessity of studying the gene cluster.

In summary, our study indicates that specific oxygenases and their auxiliary factors in the *Acinetobacter_piscicola* p38 strain play a key role in the biodegradation of indole and skatole. These findings not only enhance our understanding of the biodegradation mechanisms of these pollutants but also provide an important molecular basis for the development of new bioremediation technologies. In the future, we will continue to optimize the degradation conditions of these enzymes and explore their application potential in actual environmental management.

## 4. Conclusions

In this study, the newly acquired bacterial strain *Acinetobacter_piscicola* p38 demonstrated efficient degradation capabilities for indole and skatole under a wide range of conditions. Notably, the gene gene1687 was identified as having the ability to degrade both indole and skatole. These findings are expected to broaden our understanding of the metabolic processes and degradation mechanisms of skatole in the *Acinetobacter_piscicola* p38 strain. Elucidating the genetic basis of this strain’s robust biodegradative activity may help develop new strategies for bio-remediation of environments contaminated with these harmful compounds. Moreover, the identification and characterization of gene1687 and other related genes may pave the way for genetic engineering approaches to enhance the biodegradation efficiency of similar bacterial strains. This study not only advances our knowledge of microbial degradation pathways but also provides valuable resources for environmental protection and waste management applications.

## Figures and Tables

**Figure 1 microorganisms-12-01792-f001:**
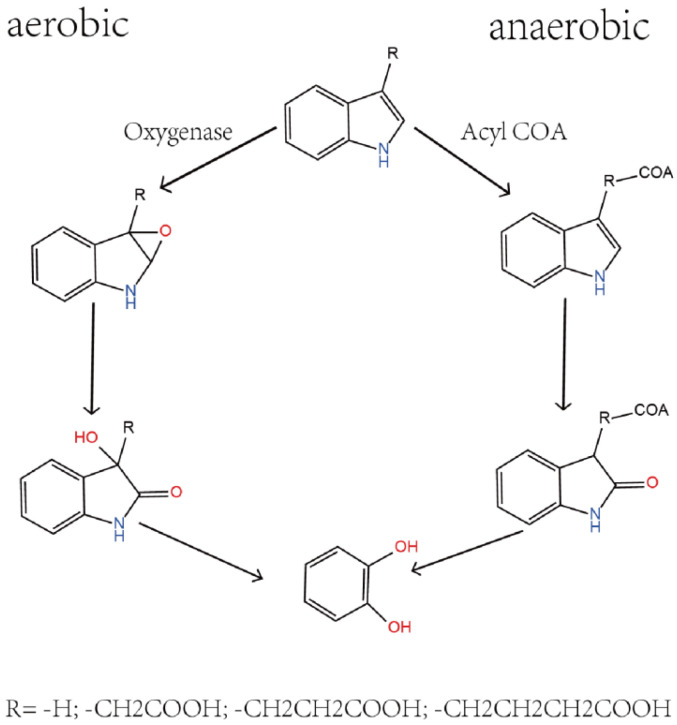
Metabolic relationships associated with indole and its derivatives.

**Figure 2 microorganisms-12-01792-f002:**
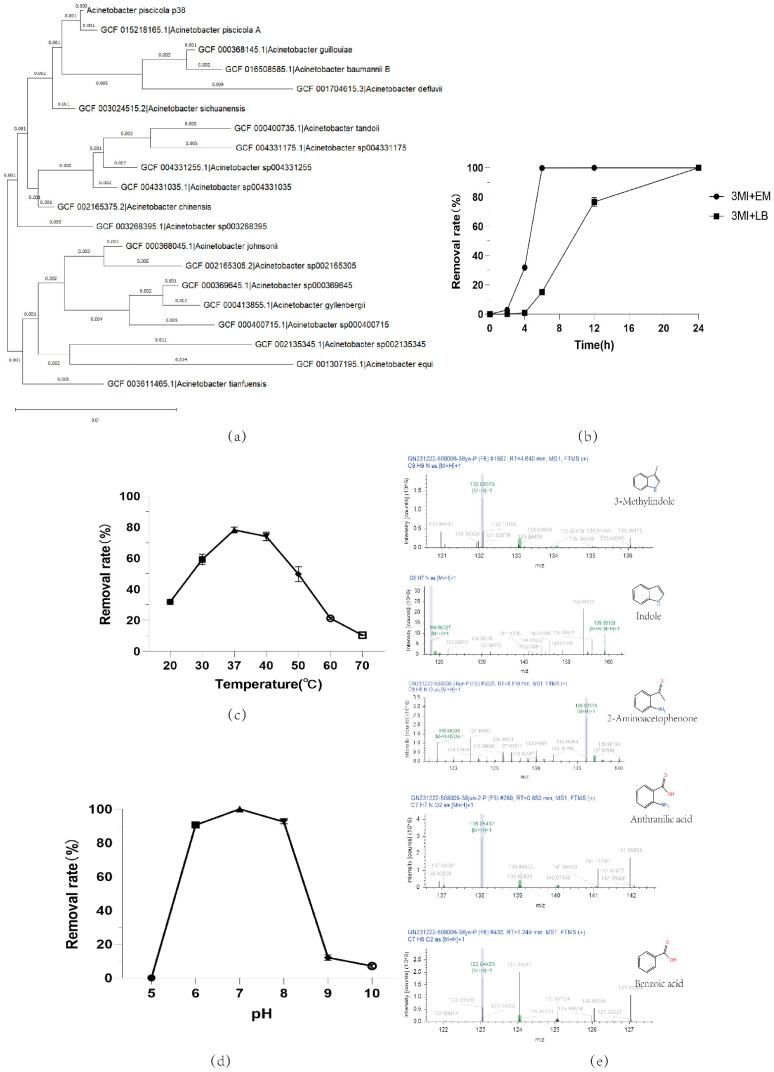
(**a**) Phylogenetic relationship of *Acinetobacter_piscicola* p38 16s RNA; skatole biodegradation by *Acinetobacter_piscicola* p38 under various growth conditions: (**b**) different cultivation times and culture media, 3MI + EM: 100 mg/L skatole + enrichment medium; 3MI + LB: 100 mg/L skatole + LB medium; (**c**) different temperatures; (**d**) different pH levels; (**e**) metabolite detection in cultures utilizing skatole as the sole carbon source.

**Figure 3 microorganisms-12-01792-f003:**
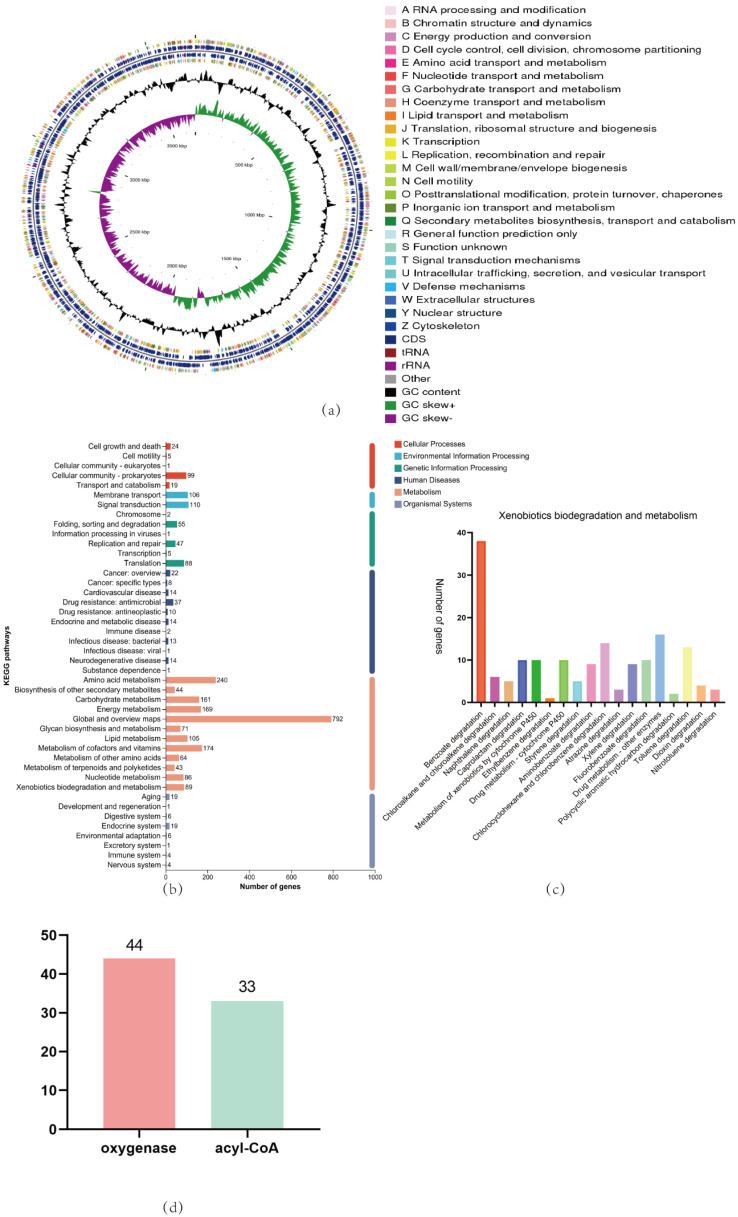
(**a**) The CGView genome circle diagram can comprehensively display the characteristics of the genome, with the information from the outermost to the inner circle corresponding to the gene information on the positive and negative strands, GC content, GC-Skew, and genome size indicator; (**b**) A detailed map of the KEGG pathways enrichment for the whole genome; (**c**) A detailed map of the KEGG pathways enrichment for Xenobiotics biodegradation and metabolism; (**d**) Analysis of oxygenases and acyl-CoA in *Acinetobacter_piscicola* p38 through whole-genome analysis.

**Figure 4 microorganisms-12-01792-f004:**
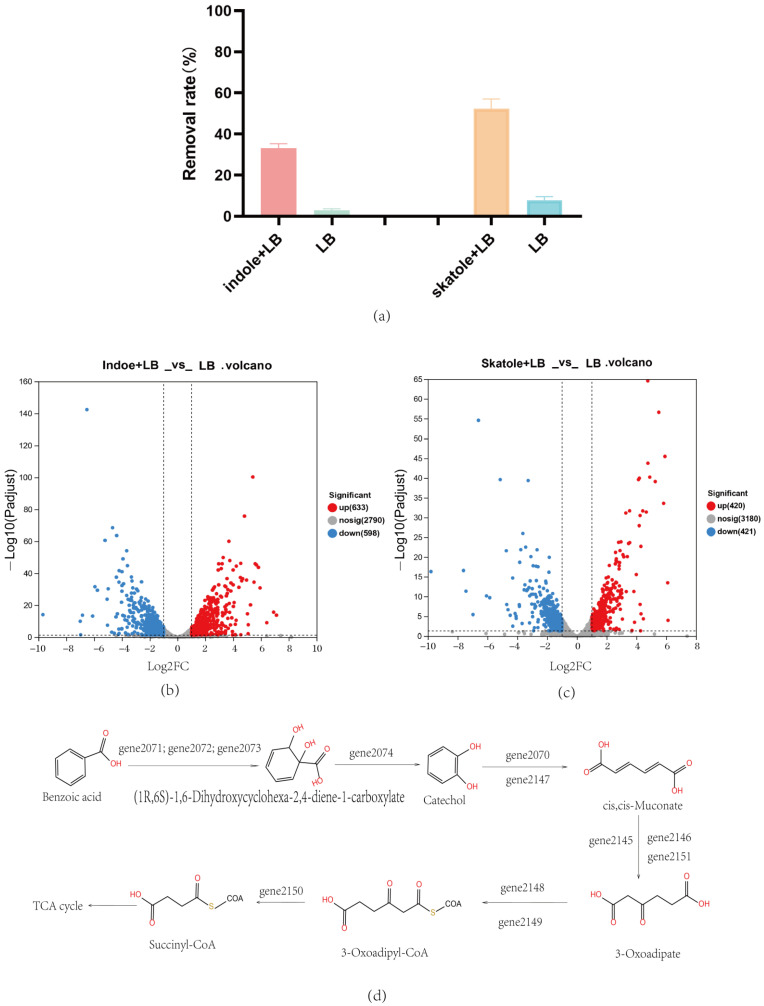
(**a**) *Acinetobacter_piscicola* p38 bacterial lysate study; (**b**) Indole + LB vs. LB: volcano plot of differential gene expression; (**c**) Skatole + LB vs. LB: volcano plot of differential gene expression; (**d**) Speculation of benzoic acid biodegradation mechanism by *Acinetobacter_piscicola* p38 through integrated analysis.

**Figure 5 microorganisms-12-01792-f005:**
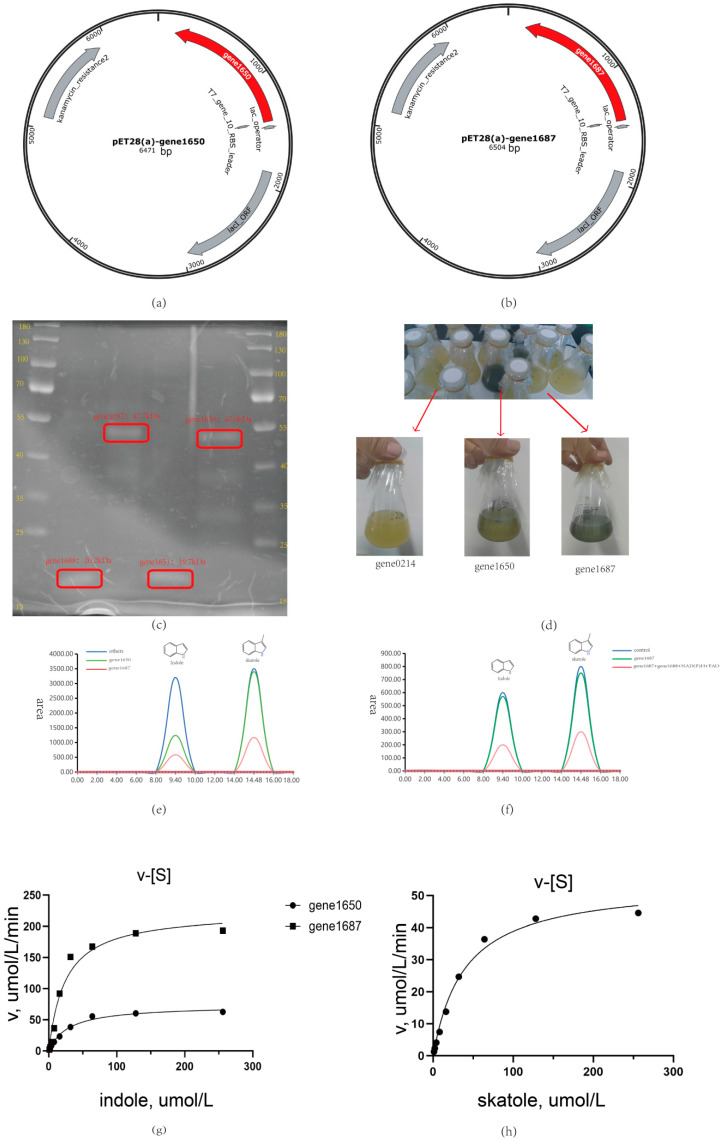
(**a**) Gene1650 plasmid; (**b**) Gene1687 plasmid; (**c**) Gene1650 and gene1687 heterologous expression coomassie brilliant blue staining; (**d**) Whole-cell biotransformation of indole; (**e**) Determination results of whole cell biotransformation by liquid chromatography; (**f**) Degradation of indole and skatole after purification of gene1687 protein; (**g**) The degradation rate of indole by gene1650 and gene1687; (**h**) The degradation rate of skatole by gene1687.

**Figure 6 microorganisms-12-01792-f006:**
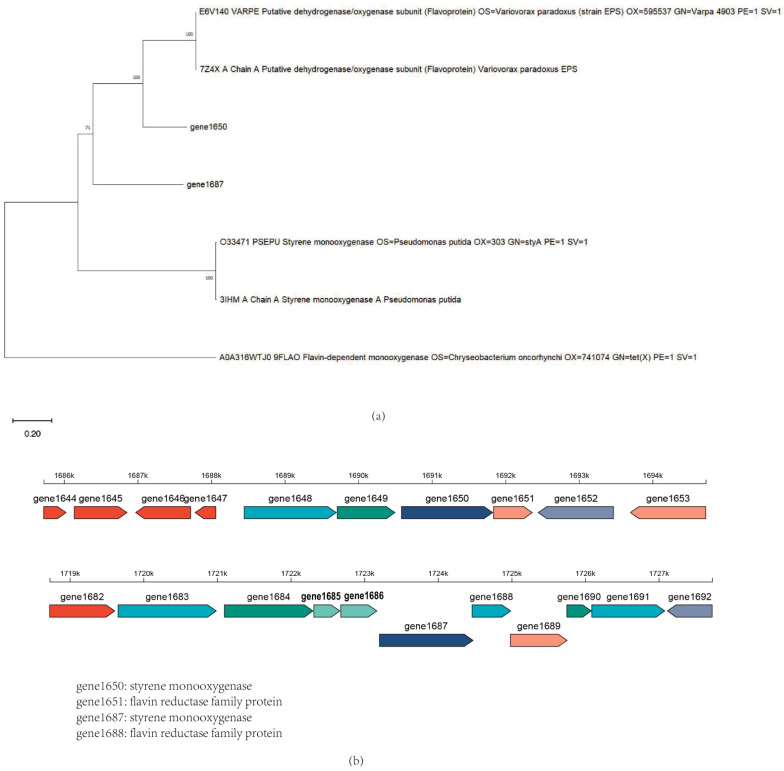
(**a**) The evolutionary tree of gene1650 and gene1687 proteins; (**b**) The gene cluster of gene1650 and gene1687.

**Table 1 microorganisms-12-01792-t001:** Screening of skatole-degrading bacteria.

Cow	Removal Rate	Chicken	Removal Rate	Sheep	Removal Rate	Pig	Removal Rate
1	18.51%	11	1.89%	21	0.52%	31	38.06%
2	0.50%	12	9.39%	22	none	32	none
3	2.47%	13	none	23	none	33	1.39%
4	19.38%	14	none	24	5.45%	34	none
5	0.76%	15	35.74%	25	none	35	100.00%
6	none	16	12.75%	26	8.79%	36	0.39%
7	none	17	86.34%	27	4.11%	37	none
8	none	18	41.84%	28	3.28%	38	100.00%
9	none	19	5.73%	29	none	39	none
10	none	20	none	30	1.60%	40	none

**Table 2 microorganisms-12-01792-t002:** Indole + LB vs. LB: significant upregulation of oxygenase and acyl-CoA in prokaryotic transcriptome.

Gene_Id	Gene Name	Gene Description
gene0215	sfnG	MULTISPECIES: dimethyl sulfone monooxygenase SfnG
gene0321	-	MULTISPECIES: DOPA 4,5-dioxygenase family protein
gene1650	styA	styrene monooxygenase
gene1684	-	aromatic ring-hydroxylating dioxygenase subunit alpha
gene2073	benC-xylZ	benzoate 1,2-dioxygenase electron transfer component BenC
gene2147	catA	MULTISPECIES: catechol 1,2-dioxygenase
gene2332	npd	MULTISPECIES: nitronate monooxygenase
gene2700	tauD	MULTISPECIES: taurine dioxygenase
gene3162	mhpB	MULTISPECIES: 3-carboxyethylcatechol 2,3-dioxygenase
gene3524	ssuD	FMNH2-dependent alkanesulfonate monooxygenase
gene0214	dszC	acyl-CoA dehydrogenase family protein
gene0703	dmdC	MULTISPECIES: acyl-CoA dehydrogenase C-terminal domain-containing protein
gene0704	dmdC	MULTISPECIES: acyl-CoA dehydrogenase C-terminal domain-containing protein
gene1225	desA3	MULTISPECIES: acyl-CoA desaturase
gene2011	bcd	MULTISPECIES: acyl-CoA dehydrogenase family protein
gene2462	ybgC	MULTISPECIES: tol-pal system-associated acyl-CoA thioesterase
gene2913	atuD	MULTISPECIES: acyl-CoA dehydrogenase family protein
gene2914	atuC	MULTISPECIES: acyl-CoA carboxylase subunit beta
gene2970	dmdB	MULTISPECIES: acyl-CoA synthetase
gene3178	atuH	MULTISPECIES: long-chain-acyl-CoA synthetase

**Table 3 microorganisms-12-01792-t003:** Skatole + LB vs. LB: significant upregulation of oxygenase and acyl-CoA in prokaryotic transcriptome.

Gene_Id	Gene Name	Gene Description
gene0215	sfnG	MULTISPECIES: dimethyl sulfone monooxygenase SfnG
gene1684	-	aromatic ring-hydroxylating dioxygenase subunit alpha
gene1687	-	MULTISPECIES: styrene monooxygenase
gene1690	hcaC	MULTISPECIES: non-heme iron oxygenase ferredoxin subunit
gene2070	catA	MULTISPECIES: catechol 1,2-dioxygenase
gene2071	benA-xylX	MULTISPECIES: benzoate 1,2-dioxygenase large subunit
gene2072	benB-xylY	MULTISPECIES: benzoate 1,2-dioxygenase small subunit
gene2073	benC-xylZ	benzoate 1,2-dioxygenase electron transfer component BenC
gene2147	catA	MULTISPECIES: catechol 1,2-dioxygenase
gene2332	npd	MULTISPECIES: nitronate monooxygenase
gene2700	tauD	MULTISPECIES: taurine dioxygenase
gene3524	ssuD	FMNH2-dependent alkanesulfonate monooxygenase
gene0214	dszC	acyl-CoA dehydrogenase family protein
gene0660	ybgC	MULTISPECIES: acyl-CoA thioesterase
gene0703	dmdC	MULTISPECIES: acyl-CoA dehydrogenase C-terminal domain-containing protein
gene0704	dmdC	MULTISPECIES: acyl-CoA dehydrogenase C-terminal domain-containing protein
gene1225	desA3	MULTISPECIES: acyl-CoA desaturase
gene1970	paaH	3-hydroxyacyl-CoA dehydrogenase
gene2011	bcd	MULTISPECIES: acyl-CoA dehydrogenase family protein
gene3178	atuH	MULTISPECIES: long-chain-acyl-CoA synthetase

## Data Availability

All detailed data are provided in the Appendix A.

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
