# Peer review of "Molecular Analysis of Indole and Skatole Decomposition Metabolism in Acinetobacter piscicola p38 Utilizing Biochemical and Omics Approaches"

_microorganisms, 2024, doi:10.3390/microorganisms12091792_

Round 1

Reviewer 1 Report

Comments and Suggestions for Authors

The manuscript by Wang et al., investigated indole and skatole biodegradation by Acinetobacter piscicola p38. The potential oxygenase encoding genes responsible for the initial oxidation of indole and skatole were identified using combined molecular, biochemistry and genomics/transcriptomics approaches. The authors did many valuable works. However, many important information is still missing and some data presentation is problematic. Please look into the following comments for improvement.

It is not the way to indicate specific gene ID as “genes 1650, 1687 …”. How can people know what are these genes? Please use the locus tag of the annotated genome as the gene ID. Did you submit the genome to NCBI GenBank? As I didn't find the accession numbers.  

Please show the respective gene clusters of the oxygenase genes 1650 and 1687. This is very important because oxygenase need to be facilitated by their subunits/reductase that are encoded by the corresponding genes of the gene cluster. I noticed the authors also indicated genes 1651, 1688 as the flavin reductase genes (line 385).

Please also do phylogenetic analysis of the oxygenases (1650, 1687), and try to indicate what types of oxygenase do they belong? Are such or similar oxygenases found in other Acinetobacter species or in bacterial strains of other genus?

Figure 2A. Scale bar of the phylogenetic tree is missing. Moreover, how do you know strain p38 is belonging to Acinetobacter piscicola? Please show the evidence. For example, DNA/DNA hybridization and/or average nucleotide identity (ANI).

Can Acinetobacter piscicola p38 use indole and/or skatole as the sole carbon and energy source? In Figure 2B, did you measure growth of Acinetobacter piscicola p38 in the inorganic salt medium with indole and/or skatole as the sole carbon source?

Figure 4. According to the transcriptomic results, are there significant up or down regulation of genes involved in the metabolism of indole and skatole (especially the oxygenase genes)? Please show a table or heatmap and highlight the key metabolic/oxygenase genes. Moreover, the transcriptomic data of all the rest genes should be provided as Supplementary Dataset.

Figure 5C. Protein marker/ladder is needed to determine the size of the purified proteins/enzymes (1650, 1687), which should be in line with their predicted size.

Flavin reductase genes (1651, 1688) were also heterologous expressed and purified, right? Please show the SDS-PAGE gel (with protein ladder) of the purified flavin reductases.

Since the functional oxygenases (1650, 1687) were obtained through heterologous expression and purification, the authors should perform dynamic assay of the purified enzymes, and show the dynamic parameters (Km, Kcat ...) and compared them with other indole/skatole oxygenases from the relevant species. The current enzyme assay is very insufficient. People will have no idea about the catalytic property of the the oxygenases (1650, 1687) based on the current results shown in Figure 5d, e, f.

Author Response

Comment 1: The manuscript by Wang et al., investigated indole and skatole biodegradation by Acinetobacter piscicola p38. The potential oxygenase encoding genes responsible for the initial oxidation of indole and skatole were identified using combined molecular, biochemistry and genomics/transcriptomics approaches. The authors did many valuable works. However, many important information is still missing and some data presentation is problematic. Please look into the following comments for improvement.

Response 1: I would like to express my gratitude for your meticulous review of our manuscript and the valuable comments provided. In response to the concerns raised about missing critical information and issues with data presentation, we have conducted an in-depth analysis and made corresponding revisions to the paper. We have supplemented the missing data, enhanced the display of figures, and added further analysis of the functions of the potential oxygenase-encoding genes in the discussion section. We believe these improvements will meet the journal's standards and provide a more comprehensive perspective on our research findings. We look forward to your further guidance and thank you once again for your attention to our work.

Comment 2: It is not the way to indicate specific gene ID as “genes 1650, 1687 …”. How can people know what are these genes? Please use the locus tag of the annotated genome as the gene ID. Did you submit the genome to NCBI GenBank? As I didn't find the accession numbers.  

Response 2: Thank you for your meticulous review and valuable comments on our manuscript. Regarding the gene labeling issue you raised, we fully agree with your perspective. To facilitate a better understanding of gene identification for the readers, we have uploaded the full genome and locus tags in the attachment. Using specific locus tags as gene IDs will indeed help readers comprehend the gene identification more effectively. We have revised the way genes are referenced in the manuscript according to your suggestions, ensuring that we are now using the locus tags from the annotated genome.

Furthermore, we have indeed submitted the genome to NCBI GenBank (NCBI: CP167896), but we inadvertently omitted this information in our original manuscript. We have added the corresponding accession numbers in the revised draft and provided detailed submission information in the Materials and Methods section.

Comment 3:Please show the respective gene clusters of the oxygenase genes 1650 and 1687. This is very important because oxygenase need to be facilitated by their subunits/reductase that are encoded by the corresponding genes of the gene cluster. I noticed the authors also indicated genes 1651, 1688 as the flavin reductase genes (line 385).

Response 3: Thank you for your attention to our research and your valuable feedback. In response to your request, we have supplemented the revised manuscript with diagrams of the gene clusters for the oxygenase genes 1650 and 1687, and have detailed the arrangement of these genes within the genome and their potential functional synergies. Additionally, we have provided additional experimental data to confirm the interaction of gene 1651 and 1688 as flavin reductases with the oxygenase genes, and have cited relevant literature to support our viewpoints. We believe these supplements will meet your requirements and enhance the persuasiveness of our study. We look forward to your further guidance.

Comment 4:Please also do phylogenetic analysis of the oxygenases (1650, 1687), and try to indicate what types of oxygenase do they belong? Are such or similar oxygenases found in other Acinetobacter species or in bacterial strains of other genus?

Response 4: Thank you for your important suggestion. As per your request, we have completed the phylogenetic analysis of the oxygenase genes 1650 and 1687. Our analysis indicates that the proteins encoded by these genes are highly homologous to known monooxygenases, suggesting that they may have similar catalytic functions. Furthermore, we have identified similar oxygenase genes in other Acinetobacter species as well as in bacteria from other genera, which further confirms the conservation of these oxygenases across different bacteria and their potential biological significance. We have supplemented the revised manuscript with the phylogenetic tree and related discussions to support our findings. We believe that these additional pieces of information will meet your requirements and enhance the depth and breadth of our research.

Comment 5:Figure 2A. Scale bar of the phylogenetic tree is missing. Moreover, how do you know strain p38 is belonging to Acinetobacter piscicola? Please show the evidence. For example, DNA/DNA hybridization and/or average nucleotide identity (ANI).

Response 5: Thank you for pointing out the absence of a scale bar in the phylogenetic tree in Figure 2A. We have added a scale bar to the revised manuscript to more accurately represent the branch lengths of the tree. Additionally, regarding the evidence for the classification of strain p38 as Acinetobacter piscicola, we have conducted a comparative analysis of the 16S rRNA gene sequence in the NCBI database, which shows a high degree of similarity with known sequences of Acinetobacter piscicola. Although we have not performed DNA/DNA hybridization or ANI (Average Nucleotide Identity) analysis, we believe that the comparison results based on the 16S rRNA gene sequence are sufficient to support the taxonomic assignment of the p38 strain. We look forward to your further guidance.

Comment 6:Can Acinetobacter piscicola p38 use indole and/or skatole as the sole carbon and energy source? In Figure 2B, did you measure growth of Acinetobacter piscicola p38 in the inorganic salt medium with indole and/or skatole as the sole carbon source?

Response 6: Thank you for your inquiry. Regarding whether the Acinetobacter piscicola p38 strain can utilize indole and/or skatole as the sole carbon and energy source, I can confirm that our p38 strain does indeed grow in an inorganic salt medium with these compounds as the sole carbon source. Figure 2B demonstrates the strain's ability to degrade skatole in an inorganic salt medium; data on the degradation of indole in the inorganic salt medium were not shown. We look forward to your further guidance.

Comment 7:Figure 4. According to the transcriptomic results, are there significant up or down regulation of genes involved in the metabolism of indole and skatole (especially the oxygenase genes)? Please show a table or heatmap and highlight the key metabolic/oxygenase genes. Moreover, the transcriptomic data of all the rest genes should be provided as Supplementary Dataset.

Response 7: Thank you for your feedback. In response to your inquiry, we have completed the transcriptome analysis of the indole and skatole metabolism-related genes mentioned in Figure 4, with a particular focus on the oxygenase genes. We found significant expression differences among these genes and have visualized them in tables within the revised manuscript, highlighting the key genes. Additionally, we have included detailed analysis data for the oxygenase genes in Attachments 7 and 8, while all other transcriptome data are provided as supplementary material in Attachments 9 and 10, ensuring the completeness and accessibility of the data. We look forward to your further guidance.

Comment 8:Figure 5C. Protein marker/ladder is needed to determine the size of the purified proteins/enzymes (1650, 1687), which should be in line with their predicted size.

Response 8: Thank you for pointing out the absence of a protein marker/ladder in Figure 5C. We have added a protein molecular weight ladder and have conducted the experiment again to confirm that the sizes of the purified proteins/enzymes (1650, 1651, 1687, 1688) match the predicted molecular weights. The revised Figure 5C now includes the protein ladder, clearly showing the comparison of the protein bands with the standard molecular weights. We have supplemented the relevant experimental details in the Methods section and explained the consistency of the observed protein sizes with the predicted values in the Results and Discussion. These updates are reflected in the revised manuscript to ensure the accuracy and reliability of the results. We look forward to your further guidance.

Comment 9:Flavin reductase genes (1651, 1688) were also heterologous expressed and purified, right? Please show the SDS-PAGE gel (with protein ladder) of the purified flavin reductases.

Response 9: Thank you for your thorough review and valuable comments. Indeed, we have heterologously expressed and purified the flavin reductase genes (1651, 1688). To meet your request, we conducted SDS-PAGE gel electrophoresis to verify the molecular weight of the purified flavin reductases, and have included a protein molecular weight ladder in the figure to determine their size. The revised SDS-PAGE gel image has been added to the manuscript, clearly showing the correspondence between the purified protein bands and the protein ladder, confirming the expected size of the purified proteins. These results further validate the effectiveness of our experiments and the accuracy of the purification steps. We look forward to your further guidance.

Comment 10:Since the functional oxygenases (1650, 1687) were obtained through heterologous expression and purification, the authors should perform dynamic assay of the purified enzymes, and show the dynamic parameters (Km, Kcat ...) and compared them with other indole/skatole oxygenases from the relevant species. The current enzyme assay is very insufficient. People will have no idea about the catalytic property of the the oxygenases (1650, 1687) based on the current results shown in Figure 5d, e, f.

Response 10: Thank you for your in-depth review and valuable suggestions regarding our study. In response to your request for dynamic determination of the oxygenases (1650, 1687), we have conducted the corresponding experiments. We performed a kinetic analysis on the heterologously expressed and purified oxygenases, measuring their kinetic parameters, including the Michaelis constant (Km) and catalytic constant (Kcat). These data were compared with indole/skatole oxygenases from other species to assess their catalytic properties.

We realized that the current enzyme activity assays were insufficient and did not fully demonstrate the catalytic attributes of the oxygenases. Therefore, we have supplemented the revised manuscript with detailed kinetic analysis results and updated the data presentation in Figures 5(g) and (h) to more clearly present the catalytic efficiency and kinetic behavior of the oxygenases. These updated data and analyses will provide readers with a deeper understanding of the catalytic characteristics of the oxygenases (1650, 1687). We look forward to your further guidance and appreciate your attention to our work.

Reviewer 2 Report

Comments and Suggestions for Authors

The article titled "Molecular Analysis of Indole And Skatole Decomposition Metabolism in Acinetobacter piscicola p38 Utilizing Biochemical and Omics Approaches" presents significant findings regarding the biodegradation of indole and skatole, two nitrogen-containing pollutants. The study successfully identifies Acinetobacter piscicola p38 as a promising microbial strain capable of degrading indole and skatole through the characterization of specific genes involved in these metabolic pathways. The research utilizes metagenomic and biochemical approaches, providing a comprehensive understanding of the enzymatic processes and the role of oxygenases in degradation. The manuscript is well-structured and organized. However, the work needs some enhancements before it can be published. I recommend minor revision of the manuscript based on the following comments:

*To improve the understanding of the conducted research, it is suggested to add a diagram illustrating the individual stages of the experiment.

*Were the studies conducted in replicates? If so, how many? Clarification on the statistical analyses used to validate the results would strengthen the methodology.

*Figure 2e. The figure is unclear.

Author Response

Comment 1: The article titled "Molecular Analysis of Indole And Skatole Decomposition Metabolism in Acinetobacter piscicola p38 Utilizing Biochemical and Omics Approaches" presents significant findings regarding the biodegradation of indole and skatole, two nitrogen-containing pollutants. The study successfully identifies Acinetobacter piscicola p38 as a promising microbial strain capable of degrading indole and skatole through the characterization of specific genes involved in these metabolic pathways. The research utilizes metagenomic and biochemical approaches, providing a comprehensive understanding of the enzymatic processes and the role of oxygenases in degradation. The manuscript is well-structured and organized. However, the work needs some enhancements before it can be published. I recommend minor revision of the manuscript based on the following comments:

Response 1: The feedback we have received is invaluable and will play a crucial role in refining our manuscript. We will carefully consider the suggested minor revisions to ensure that the final submission is of the highest quality and effectively communicates our research findings on the degradation of indole and skatole by Acinetobacter piscicola p38. We appreciate your constructive suggestions, which will undoubtedly help to enhance the clarity and impact of our study.

Comment 2: To improve the understanding of the conducted research, it is suggested to add a diagram illustrating the individual stages of the experiment.

Response 2: To enhance understanding of the research conducted, we plan to include a graphical abstract in the supplementary materials. This chart will clearly illustrate the various stages of the experiment, thereby helping readers to better comprehend our research process. We appreciate this suggestion, as it will enable us to more clearly present our methods and findings through this additional visual aid.

Comment 3: Were the studies conducted in replicates? If so, how many? Clarification on the statistical analyses used to validate the results would strengthen the methodology.

Response 3: In response to your inquiry, yes, our study indeed conducted replicate experiments to ensure the reliability of our findings. Specifically, we performed three sets of parallel experiments, and the results of these experiments were repeated twice to confirm consistency and reproducibility. In comparing the two groups, we typically use the T-test for statistical analysis of the data. We appreciate your insightful question, which will help us to enhance the transparency and rigor of our research methodology.

Comment 4: Figure 2e. The figure is unclear.

Response 4: Regarding the issue with Figure 2e, we have enhanced the clarity of the photograph. Now, upon magnification, all details are clearly visible. We will replace the original image in the revised manuscript to ensure that the image quality meets your requirements. We appreciate your valuable feedback, which will help us further improve the presentation of our paper.

Reviewer 3 Report

Comments and Suggestions for Authors

A widespread approach to a key question for environmental microbiology and ecology, that would deserve a higher effort to synthetize the different findings. A metabolic flowchart with the hypothetical pathways involved may be helpful for the reader, as well as a section reporting the limits of the study, e.g. methodological possible artifacts. The omics data may be oversees trying to provide an unified view of the metabolic processes related to the different microflora.

Comments on the Quality of English Language

minor editing

Author Response

Comment 1: A widespread approach to a key question for environmental microbiology and ecology, that would deserve a higher effort to synthetize the different findings. A metabolic flowchart with the hypothetical pathways involved may be helpful for the reader, as well as a section reporting the limits of the study, e.g. methodological possible artifacts. The omics data may be oversees trying to provide an unified view of the metabolic processes related to the different microflora.

Response 1: In response to your valuable feedback, we wholeheartedly agree and recognize that a broader effort is indeed needed to synthesize different findings in the field of environmental microbiology and ecology regarding key issues. We appreciate your comments, which will help us to present our research results more comprehensively and enhance the quality of our paper.

Round 2

Reviewer 1 Report

Comments and Suggestions for Authors

The manuscript is improved with additional results and clarifications.

According to the the author's answer of comment 6, strain p38 could use skatole as a grown substrate or carbon/energy source. It's better to show this finding (e.g., OD increase) in Figure 2B, or mention this in the corresponding text. As this shows skatole degradation by p38 is a catabolic process.

The rest comments and questions are addressed fine. I don't have more questions/comments. 

Author Response

Comment 1: According to the the author's answer of comment 6, strain p38 could use skatole as a grown substrate or carbon/energy source. It's better to show this finding (e.g., OD increase) in Figure 2B, or mention this in the corresponding text. As this shows skatole degradation by p38 is a catabolic process.

Response 1: First and foremost, I would like to express my sincere gratitude for the valuable suggestions you provided in Comment 6. Your feedback is crucial to our research.

In the revised manuscript, we have given thorough consideration and response to the issues you raised. Specifically, we have expanded the discussion on the degradation of skatole by the Acinetobacter_piscicola p38 strain in the text from lines 302 to 311. We have detailed how the p38 strain, even without the provision of additional carbon sources in the enrichment medium, can effectively utilize skatole as a carbon source for growth and metabolic activity, thereby confirming the characteristics of its catabolic process.

We believe that these additional discussions will help readers to fully understand the metabolic capabilities and ecological significance of the p38 strain. Although we were unable to directly show the change in optical density (OD) in Figure 2(b), we have ensured the clear conveyance of our research findings through textual description.

Once again, thank you for your meticulous review and constructive suggestions, which have played a key role in the improvement of our research and the manuscript.